# Dynamic Duos? Jamaican Fruit Bats (*Artibeus jamaicensis*) Do Not Show Prosocial Behavior in a Release Paradigm

**DOI:** 10.3390/bs6040025

**Published:** 2016-11-20

**Authors:** Eric Hoffmaster, Jennifer Vonk

**Affiliations:** Department of Psychology, Oakland University, 2200 N Squirrel Rd, Rochester, MI 48309, USA; elhoffmaster@oakland.edu

**Keywords:** Jamaican fruit bats, prosocial, escape, recipient

## Abstract

Once thought to be uniquely human, prosocial behavior has been observed in a number of species, including vampire bats that engage in costly food-sharing. Another social chiropteran, Jamaican fruit bats (*Artibeus jamaicensis*), have been observed to engage in cooperative mate guarding, and thus might be expected to display prosocial behavior as well. However, frugivory and hematophagy diets may impose different selection pressures on prosocial preferences, given that prosocial preferences may depend upon cognitive abilities selected by different ecological constraints. Thus, we assessed whether Jamaican fruit bats would assist a conspecific in an escape paradigm in which a donor could opt to release a recipient from an enclosure. The test apparatus contained two compartments—one of which was equipped with a sensor that, once triggered, released the trap door of the adjacent compartment. Sixty-six exhaustive pairs of 12 bats were tested, with each bat in each role, twice when the recipient was present and twice when absent. Bats decreased their behavior of releasing the trapdoor in both conditions over time, decreasing the behavior slightly more rapidly in the recipient absent condition. Bats did not release the door more often when recipients were present, regardless of the recipient; thus, there was no clear evidence of prosocial behavior.

## 1. Introduction

One of the most enthusiastically studied questions in the field of comparative psychology over the last decade is the question of whether any non-human species share the human tendency to behave prosocially when there is no gain to the self. Prosocial behavior is defined as a behavior performed by one individual to improve the welfare of another individual [1]. Prosocial behavior may also be considered altruistic if the assisting individual (hereafter, donor) incurs a cost or receives no additional benefit for assisting the recipient. That is, prosocial acts toward kin are not considered truly altruistic because assisting kin adds to one’s own reproductive fitness. Likewise, reciprocal acts may also be considered to result in a net gain for the actor. A famous example of prosocial behavior in non-humans is blood sharing among common vampire bats (*Desmodus rotundus*), in which individuals who have fed will regurgitate a portion of their blood meal to feed group-mates who have not fed [2]. In the case of blood sharing, the donor takes on a high cost of losing a portion of the meal that it expended valuable time and energy to obtain, while the recipient did not expend any energy but still receives the benefit of food. 

Vampire bats are likely not the only bat species to engage in prosocial or cooperative behaviors. Darling’s horseshoe bats (*Rhinolophus darling*), a small insectivorous bat, exhibit a behavior known as group augmentation, in which offspring of the group huddle together in cave crèches so that they can stay warm through the thermoregulation of the group [3]. Carter and Wilkinson [3] suggest that, by mothers cooperating to keep all of the offspring warm, they increase the probability of their own offspring’s survival. Although the behavior cannot be considered truly prosocial, and may be better classified as mutualism or cooperation because the benefit may outweigh the cost to the donor, the presence of cooperative behaviors highlights the fact that bats will behave to assist others.

Another instance of female bats cooperating together to ensure the survival of the group offspring is pup guarding, which can be observed in greater spear-nosed bats (*Phyllostomus hastatus*), a small omnivorous bat [4]. Researchers observed that, when pups fell from the cave ceiling to the floor, adult bats from different social groups would bite and attack the infant bats, while adult females from the infants’ social group would fend off rival females to protect the infants. One possible benefit of this behavior is group augmentation and the optimization of pup numbers such that a sufficient amount of body warmth is generated to warm all pups in the group [3]. This hypothesis is supported by the fact that *Phyllostomus hastatus* groups have synchronized births [4,5].

Although strictly prosocially motivated behavior has not yet been reported in fruit bats, the Jamaican fruit bat, a small echolocating frugivorous bat, has also displayed cooperative behaviors that indicate the propensity to engage in behavior that helps another. For example, subordinate bats will mate guard by helping the dominant male of the group chase off non-related rival males [6]. Ortega and colleagues propose that both the subordinate and dominant males receive indirect fitness benefits by participating in the cooperative behavior. By helping the dominant males ward off rival bats, subordinates have greater access to the females. If related, both the dominant and subordinate bats receive indirect fitness benefits by sharing access to females [7]. Although the dominant may incur reductions in direct fitness through sharing access to females, this cost may be outweighed by the benefits of assisted mate-guarding where access to the female would otherwise be lost to rival, unrelated males.

Despite this evidence of bats participating in behaviors that benefit others in the wild, little research has provided bats the opportunity to behave prosocially in a controlled experimental design in order to tease apart the conditions under which these prosocial behaviors might occur, and to provide direct comparison to data obtained in other species, such as bonobos [8] and rats [9]. To date, a study conducted by Carter and Wilkinson [10] examining the predictors of blood sharing in vampire bats, is the only one to examine prosocial behaviors experimentally, but used a procedure very specific to the unique, natural behavior of vampire bats. 

The purpose of the current study was to examine if Jamaican fruit bats would display evidence of prosocial behavior when presented with a task in which group-mates were confined and separated from the home group. Bats could assist one another in escaping confinement to rejoin their conspecifics, and regain access to food. Because bats can form close social bonds [3,11,12] across multiple dimensions, we also aimed to examine whether bats were preferentially prosocial based on factors such as age, reproductive status, reciprocity, and kinship. 

The current study utilized a research design similar to that of Vonk et al. [13], which contrasted the frequency of ostensibly prosocial behavior in two conditions, one with a recipient present (RP) and the other condition with the recipient absent (RA). The recipient absent condition served as a control condition allowing us to differentiate between differing motivations for releasing the trap door. Only if the motivations to release the partner are prosocial, rather than selfish, should we see different frequencies and latencies of release in the RP versus RA conditions. Because bat species, such as the common vampire bat, have demonstrated prosocial behavior in the form of blood sharing [14], we hypothesized that the Jamaican fruit bat would display prosocial behavior in the current context as well. If bats behave prosocially toward conspecifics, then the number of instances in which a donor triggers the release mechanism to allow its partner to escape should be significantly greater in the condition in which the recipient is present versus when the recipient is absent. Additionally, a donor’s latency to trigger the release mechanism in the condition in which a recipient is present should be significantly faster than the condition in which the recipient is absent. It was expected that the bats would initially trigger the release mechanism at high rates in both conditions as they have learned to escape themselves, but, when they realize that they can no longer escape the box, their attempts to release the door should decline more rapidly when the recipient is absent versus present if they have prosocial motivations. If the bats do not understand the connection between their behavior and the releasing mechanism, or if they do not display prosocial motivations, than their frequency to trigger the releasing mechanism should remain similar in both conditions. 

## 2. Materials and Methods

### 2.1. Subjects

Subjects consisted of 14 adult female Jamaican fruit bats (*Artibeus jamaicensis*) housed at the Organization for Bat Conservation (OBC; Bloomfield Hills, MI, USA). Bats were segregated by sex and housed in flight cages that allowed them to freely associate with other members of their own and other species. Our sample contained two mother daughter pairings. Bat F092 was the mother of bat 346B and bat AF86 was the mother of bat 2734. No other bats in our sample were related along the matrilineal line. These subjects shared a flight cage with female short tailed fruit bats (*Carollia perspicillata*), another frugivorous bat. All animals were either micro-chipped with passive integrated transponder (PIT tags) that could be scanned by the researcher to determine their identity, or were banded on their wing, which could also be used to determine their identity. The study was approved by the Institutional Animal Care and Use Committee of Oakland University, Approval #15093.

### 2.2. Materials

The bats were exposed to an aluminum framed and meshed box apparatus that measured 61 × 30 × 30 cm (see Figure 1). The apparatus consisted of two compartments (referred to as compartment A and compartment B) and each compartment was 30 × 30 × 30 cm. Separating the two compartments was a mesh door that could be either locked or unlocked using a hook and eye lock. The floor of each compartment consisted of a door made of Lexan that could be locked and unlocked by utilizing a linear actuator. In order to retract the actuator and release the trap door, a bat had to cross an electronic sensor that was mounted in the back corner of compartment A of the apparatus. The beam of the sensor was 10.16 cm long, and when the bat crossed the sensor, an electronic signal was sent to the actuator to retract and release the trap door. The doors could also be tripped or reset by pressing two buttons connected to the electronic components of the apparatus, which were located on the side of the apparatus. By tripping the sensor in compartment A, the door for compartment B was released. Because there was no sensor in compartment B, the trap door for compartment A could be opened only manually. In this manner, a donor bat would have to help a recipient bat escape at no immediate benefit to itself, with the minimal cost of making the movement to release the recipient.

A Hero 4 Go Pro was affixed adjacent to the apparatus and used to film all trials. Additionally, a stopwatch was used to record all latencies.

### 2.3. Procedure

#### 2.3.1. Habituation

During habituation, the bats were simply allowed to investigate the apparatus by flying around and inside of it. At no point were they confined in the apparatus. The apparatus was suspended in the back corner of the flight room overnight to allow the bats to have access to the apparatus for the maximum amount of time. The bottom trap doors were open and the sensors turned off. This phase lasted until all signs of fear and anxiety around the apparatus from the bats had dissipated. 

#### 2.3.2. Phase One (Training)

The apparatus was placed in the back corner of the home cage such that it backed up against the back wall of the home cage. The inner door connecting compartments A and B to one another was unlocked and open, so that a bat could learn to activate the sensor to open the door to the adjacent compartment. Because it would have access to the adjacent compartment during this phase, the bat should have been motivated to activate the sensor and open the door, which was the only way that it could escape into the larger home enclosure. A single bat was placed in compartment A of the apparatus for five minutes. After five minutes, if the bat was unable to escape, a staff member manually released the bat by opening the door in the adjacent chamber and allowing the bat to return to its home group. All bats participated in one training session per test day, which occurred three times per week. The order in which bats were placed in the apparatus on each test day was randomly determined. 

All trials were recorded on video. During trials, a researcher recorded the bat’s identity, if the bat successfully opened the trap door, the latency to activate the trap door, and if the bat was successfully able to exit the apparatus (along with the latency to exit). Those bats that were able to exit the apparatus on two consecutive trials moved on to Phase Two. Training continued until 12 bats reached criterion. 

#### 2.3.3. Phase Two (Testing)

All possible dyads of the 12 bats that met criteria (N = 66 dyads) were systematically paired for four trials (two RA and two RP) within the apparatus. Dyads were tested in random order; however, no bat was tested in more than one dyad per day. Bats within a dyad participated as both donor and recipient on alternating trials on the same test day with recipient present (RP) and recipient absent (RA) trials alternating on consecutive test days.

During RP conditions, both a donor and recipient were present, each in different compartments, whereas, during the RA condition, only the donor was present in compartment A. The RA condition served to test whether the prosocial response occurred specifically when a partner could benefit, and extinguished faster when there was no partner to benefit. 

All trials were recorded on video. Similar to Phase One, the apparatus was suspended in the back corner of the home cage, against the back wall of the home cage. Unlike in Phase One, the inner mesh door was closed and locked. On RP trials, one member of the dyad was placed in compartment A, while the other member of the dyad was placed in compartment B of the apparatus. The left compartment (compartment A) always contained the sensor and contained the donor. In order to escape the apparatus and rejoin the home group, a donor bat needed to assist a recipient bat by hanging in the correct area of the compartment to trip the sensor. By doing so, the donor bat was able to release the trap door in their recipient’s compartment, allowing the recipient to escape.

On each trial during the RP condition, a single dyad was placed in the testing apparatus. Two trials per day were conducted with each dyad being tested for five minutes per trial. Trials always lasted five minutes even if the recipient bat was released. This was done to ensure that there was no incentive for the donor to release the trap door faster so that it could rejoin the colony to satisfy its own selfish motivations. If the donor bat failed to release the recipient bat by the end of the trial, a staff member manually released the bat(s) and returned them to the home group. During each trial a researcher would record, if the donor released the trap door for the recipient and the latency for the actor to release the trap door. 

The RA condition was identical in all respects to the RP condition, with the exception that there was no recipient bat in the adjacent chamber. During the RA condition, a researcher recorded if a bat released the trap door, and if so the researcher recorded the latency for the bat to release the trap door.

After testing all dyads once, it was observed that some of the subjects were beginning to display signs of becoming unmotivated to move within the apparatus such as hanging motionless near the inner mesh door of the apparatus for a majority of the trial. To combat this, the dimensions of compartment B were shrunk by 8 cm on each side to provide a slightly more stressful state for the recipient, and to prevent the actor from hanging on the side of the mesh immediately adjacent to the recipient. Researchers examining prosocial behavior in rats using an escape paradigm [9] constrained the recipient in a small compartment in order to cause the recipient distress that the donor could potentially pick up on. Subsequently, Sato et al. [15] placed a distressed recipient rat into a small compartment filled with water, which the donor rat could opt to release the recipient rat from into a safe area. By creating a more stressful situation for the recipient bats in the current study, donor bats may also detect their distressed state and be more motivated to participate and act prosocially. Moreover, by making compartment B smaller, it would provide another cue to signal to the bats whether they were participating in the role of donor or recipient on a given trial. In addition, as some bats were hanging near the inner mesh door near the recipient, making compartment B smaller added an extra 8cm of separation between the bats and encouraged the donor to move around the compartment rather than hang near the recipient. Lastly, in the second block of trials a small cloth was hung on the side of the sensor that the bat had to go under or around in order to activate the sensor. The addition of the cloth further assisted the bat in determining when it was in the donor role versus the recipient role. After the modification was completed, all dyads were retested. 

Because of the modification to the apparatus, we considered the test phase to consist of two distinct phases. Block one consisted of the trials that occurred before the modification took place, and Block two consisted of trials that occurred after the modification took place. Over the course of both time blocks and the entirety of testing, each bat was paired with each recipient once in each test block, once as donor and once as recipient. In the second block of testing, bat 346B was pregnant and gave birth to a pup. Bat 346B was then removed from the study so as not to add stress on her and the pup. To ensure that each bat was able to participate as a donor each test session, one bat was randomly selected each night to participate twice as a recipient to fill the gap left by her absence. Other than the modification of the apparatus and one bat being randomly assigned to participate twice as a recipient each test session, there were no procedural differences between the two blocks of testing. 

### 2.4. Statistical Analyses

Two repeated measures ANOVAs were conducted. The first assessed whether there was a significant difference in the average frequency of the bats to release the trap door of compartment B between time blocks and test conditions. The second assessed whether there was a significant difference in the average latency of the bats to release the trap door of compartment B between time blocks and test conditions. We predicted an interaction such that frequency of release should decline across blocks, but should decline more rapidly in the RA condition compared to the RP condition. Secondly, we predicted an interaction that latency of release should decline across blocks, but should decline more rapidly in the RP condition compared to the RA condition. Because there were several trials on which the bats did not release the trap door, we assigned a maximum trial length to the latency variable on these trials. However, this led to the data not being normally distributed. We thus also conducted paired-*t*-tests to compare the latency of release on RP and RA trials for only the subset of trials on which the trap door was released.

Two additional repeated measures ANOVAs were then performed in order to assess if bats as donors were more likely to assist any specific recipient bats compared to other recipient bats. First, a Repeated Measures ANOVA of each donor bats’ average frequency to release the trap door of compartment B for each recipient bat was conducted. Secondly, a Repeated Measures ANOVA of each donor bat’s average latency to release the trap door of compartment B for each recipient bat was conducted. To account for instances of missing data, as donors could not be paired with themselves as recipients, the total average frequency and total average latency of the donor to release compartment B was used in these cells. We predicted a main effect of recipient such that frequency and latency to release the trap door would vary as a function of the identity of the bats.

Individual goodness of fit chi-square tests were conducted for each of the 12 subjects to determine if any individual bats behaved prosocially regardless of whether the overall effects were significant. Chi-square tests compared the percentage of trials in which each donor bat activated the sensor to release the trap door when a recipient bat was present and when a recipient bat was absent. We predicted that prosocially motivated bats would have a significantly higher percentage of trials in which they released the trap door when a recipient bat was present relative to when a recipient bat was absent. 

## 3. Results

### Reliability

An independent rater coded 47% of the trials (both RA and RP trials). A Pearson correlation was conducted on the trials coded by the primary and secondary raters, and the coders showed strong agreement, (*N* = 248, *r* = 0.985, *p* < 0.001).

Two repeated measures ANOVAs were conducted to assess whether there was a significant difference in frequency of the bats to release the trap door of compartment B between time blocks and test conditions, and secondly a significant difference in latency of the bats to release the trap door of compartment B between time blocks and test conditions. Two factors were analyzed; test conditions (RP, RA), and time blocks (time block one, time block two). There was no significant effect of testing condition on frequency, *F*(1,11) = 1.986, *p* = 0.186, η^2^ = 0.153, with an observed power of 0.251, but there was a significant effect of time block *F*(1,11) = 5.069, *p* = 0.046, η^2^ = 0.315, with an observed power of 0.537. The bats generally released the trap door less often over time. In addition, the interaction between time and condition approached significance *F*(1,11) = 3.771, *p* = 0.078, η^2^ = 0.255, with an observed power of 0.426. Figure 2 represents the bats’ average frequency to release the trap door of compartment B for each testing condition for each time block. Examining Figure 2, it can be seen that, while the bats’ rate of releasing the trap door when a recipient was present decreased between time blocks, the frequency of releasing the door decreased more drastically when a recipient was absent. This finding, although only a trend, was consistent with predictions. However, as can be seen in Figure 2, the more drastic decline in the RA condition was not the result of behavior falling to lower rates than in the RP condition, but rather, due to a higher level of performing the behavior in block 1 when the recipient was absent.

There were no significant effects of condition or time on latency to release the door and no significant interaction.

Two additional repeated measures ANOVAs were then performed in order to assess if donors were more likely to assist any of the other specific bats as a recipient. The within subjects factor was the identity of the paired recipient (*N* = 12). The first ANOVA assessed the donor’s average frequency to release the trap door of compartment B for each recipient and revealed no significant effects of recipient *F*(1,11) = 0.842, *p* = 0.378, η^2^ = 0.071 with an observed power of 0.134, suggesting that donor bats did not release the trap doors differentially depending upon the recipients. The second ANOVA assessed the donor’s average latency to release the trap door of compartment B for each recipient and also revealed no significant differences *F*(1,11) = 0.002, *p* = 0.965, η^2^ = 0.000, with an observed power of 0.050 suggesting that donor bats did not release the trap doors at faster rates for any particular recipients. It is possible that with increased power and more trials with each recipient, an effect would have been more likely to be obtained.

In addition, we assessed whether any individual’s frequency to release the trap door of compartment B significantly differed between RP and RA conditions by conducting 12 individual Goodness of fit Chi-Square tests. Figure 3 displays the frequency for each bat to release the trap door of compartment B in each test condition and trial block across all recipient pairings (*N* = 11). Only one test, for bat 1OE7, revealed a significant difference *χ*^2^(1, *N* = 22) = 13.20, *p* = 0.001 between RP and RA conditions. Contrary to predictions, this bat released the trap door significantly more often on the RA (*N* = 17) than on RP (*N* = 4) trials. It should be noted that A2A7 also showed a trend approaching significance to release the trap door of the recipient’s compartment more on RA (*N* = 18) than RP (*N* = 11) trials; *χ*^2^(1, *N* = 22) = 4.956, *p* = 0.055. Although bat COD3 released the trap door of compartment B more often in the RP (*N* = 16) condition than in the RA (*N* = 11) condition, this difference was not significant. Thus, contrary to predictions, none of the individual bats behaved in a manner consistent with prosocial motivations. 

Because of the number of trials in which a donor did not release the trap door, the latency data was not normally distributed when a maximum trial duration was assigned as the value for latency. There is not an appropriate transformation for such data. Thus, we opted to omit trials on which the trap door was not released and re-analyze this subset of data. When viewed using a histogram, these data did appear to be normally distributed. Thus, we also conducted a paired *t*-test comparing latency to release on RP and RA trials including only trials on which the door was released. The difference approached significance, with bats showing a very small tendency to release the door more quickly when recipients were present (M = 99 s) compared to when they were absent (M = 114 s), *t*_141_ = −1.70, *p* = 0.09).

## 4. Discussion

The results from this study indicated that, when bats were separated from the group and placed in two compartments where one bat could display prosocial behavior by releasing a conspecific, the bats failed to demonstrate evidence of prosocial motivations either at the group or individual level. Furthermore, although results indicated that, over time, the bats exhibited a sharper decline of release in the RA condition as hypothesized, the overall rate of release in the second block of testing was similar for RA and RP conditions. Thus, there is no clear evidence from this study that Jamaican fruit bat females are motivated to perform a behavior to release conspecifics that cannot release themselves.

Bats 1OE7 and A2A7 released the trap door more often in the RA condition than in the RP condition. Although these two bats did release the trap door more often in the RA condition, when examining their individual data in Figure 3, it can be observed that these bats performed differently between the two time blocks of testing. Bat A2A7 had one of the sharpest decreases in releasing the trap door in the RA condition between time blocks where she went from releasing the trap door 10 times in the first time block to releasing the trapdoor only three times in the second time block of the RA condition. What is interesting is that such a sharp decline was not observed in the RP condition between trials where bat A2A7 released the trap door six times in the first time block, to releasing the trap door five times in the second time block. Examining Bat 1OE7’s data, a slight decrease in the frequency of releasing the trap door in the RA condition is observed between time blocks by one trial; however, interestingly, bat 1OE7 went from releasing the trap door zero times in the RP condition during the first time block to releasing the trap door four times during the second time block. Thus both bats display variation in their frequencies to release the trap door in both test conditions between time blocks—possibly indicating that the bats realized they could not escape from the apparatus themselves. It is thus possible that bats may have exhibited learned helplessness where they gave up trying to release themselves and decreased their behavior to trigger the releasing mechanism in time block 2. Additionally, the variation in the bats’ frequencies to release the trap door in both test conditions may indicate why bats 1OE7 and A2A7 were observed to release the trap door more often in the RA condition from block 1 to block 2, whereas, as a group, the bats were observed to significantly release the trap door less in the RA condition from block 1 to block 2.

To explain the difference in frequencies to release the trap door in the RA condition between time blocks and the observed trend, it is possible that reducing the size of compartment B in the second time block may have had an influence on the donor’s behavior. As the recipient rats in [9] displayed signs of distress in a small compartment, the recipient bats in the current study may have become distressed after the compartment was made smaller. In a distressed state, the recipient bats may have begun to produce ultrasonic vocalizations. The donor bats could have detected the vocalizations, providing an extra cue as to when a distressed recipient was present or absent. The donor bats then could continue to release the trap door in the RP condition at a similar frequency between both trial blocks in order to respond to the vocalization and then decrease releasing the trap door in the RA condition because there is no vocalization to respond to. As ultrasonic vocalizations were not recorded in the present study, this is an aspect that future research should address.

The lack of finding a prosocial behavior in the Jamaican fruit bats is somewhat surprising but also consistent with other lab studies examining the presence of prosocial behavior in other non-human animals. Studies of prosocial behavior in chimpanzees have both failed to demonstrate evidence of prosocial behavior [13,16] and provided evidence for prosocial behavior [1,17]. Evidence of prosocial behavior is mixed in other species including capuchins [18,19] and cooperative breeders, such as tamarins [20,21]. As this is the first study to examine prosocial behavior in Jamaican fruit bats, it is imperative that further research be conducted as this single lack of finding of prosocial behavior in this study’s specific context does not indicate that the species lacks the capacity for prosocial motivations in other contexts. Further research examining prosocial behavior in bats may yield positive results, building upon the methodology of the current study and overcoming some of the limitations. 

Our experiment hinged upon the comparison of release rates and latencies between RP and RA conditions. If the bats had behaved differently in these two conditions, we could have ruled out the possibility that they failed to associate their own behavior (moving near the sensor) and the outcome of releasing a conspecific. However, given the lack of main effect of test condition, it remains possible that bats did not form that association, thus making it impossible for them to reveal prosocial motivations.

Another significant limitation of the current study is that each bat interacted with each recipient only twice throughout each phase of the experiment. More trials might be needed to detect interactions between individual actors and recipients. Another limitation was the size that compartment B could be reduced to during the second time block of testing. In [9], the recipient rats were constrained in a manner that may have caused high levels of distress that donor rats could presumably detect, which might have motivated them to release their trapped cage mates. Whereas reducing the size of the recipient compartment in the current study may have had an influence on how the bats reacted during the second time block, it is possible that if the compartment was reduced to a size that would immobilize a recipient bat, these bats would display higher levels of distress that donor bats could detect and which might be more likely to motivate them to engage in prosocial behavior. However, in the current study, it was not desirable to reduce the size of the recipient compartment to cause added stress on the recipients. Even if this had been possible, donor bats might have behaved prosocially in order to reduce their own distress in hearing the distress calls of their conspecifics. A behavior motivated by the need to reduce one’s own distress would not constitute evidence for prosocial motivations, which are defined by the concern regarding another’s distress. The literature has been fraught with challenges in dissociating prosocial behaviors from their underlying motivations, and this remains a challenge [22]. 

A final limitation that the current study encountered was successfully motivating the bats to participate during the data collection. A series of pilot procedures utilizing food as a motivational reward, to test for the presence of prosocial behavior in the bats indicated that the bats were uninterested in the food reward and unwilling to participate. As Jamaican fruit bats live in stable groups of several females with one or two males and spend a significant amount of time in close contact [23], the current study then used an escape paradigm to test for prosocial behavior in females. Although this seemed to be a promising method that future research could build upon, it is possible that the mechanism to release the trap door may have been too easy—contributing to reasonably high activation rates in the RA condition, which may have masked the ability to detect a difference between release rates in that condition and the RP condition. A cloth was placed in front of the sensor to add a level of difficulty in that the bats would have to go under or around the cloth in order to activate the sensor. However, the addition of the cloth may not have sufficiently increased the difficulty of the task and future research may implement a solid barrier on multiple sides of the sensor making the task more difficult for the bats. Additionally, it was difficult to detect behavior as some bats remained still for the majority of their testing sessions. One such bat that remained still for a majority of her trials was bat 7D4B who participated on only four trials in the RP condition and seven trials in the RA condition. The bats’ frequency to release the trap door in the RP condition ranged from 0 to 22 instances and from 5 to 22 instances in the RA condition. If a bat failed to release the trap door, the latency recorded was five minutes indicating that the sensor had not been activated for the duration of the trial. This decision rule would have artificially deflated the response time to release a conspecific for bats that did not release at all during the maximum trial length. However, we also measured frequency as a dichotomous variable and the same lack of effects were detected using this measure. Furthermore, omitting trials where the door was not released from the analysis led to a very small difference in release latencies between conditions.

It is possible that prosocial behavior in bats is exhibited only by vampire bats. Vampire bats feed in a unique way and have to feed at regular intervals or they will expire [2]. Due to their unique dietary requirements, behaviors that facilitated strong social bonds, such as reciprocity, which has been found to be an important predictor of blood sharing, may have been naturally selected and evolved within the species [10]. Among bats, vampire bats have among the largest relative brain size [24]. The combination of a large brain size along with a unique feeding strategy in a species that needs to feed every day may have aided in the facilitation for the evolution of prosocial behavior in these species. Those bats that were more prosocial helped other prosocial bats by sharing blood meals and, hence, survived to pass on their genes, while bats that were not recipients of prosocial acts would not have survived to reproduce, thus increasing the rate of prosociality within the vampire bat population. 

It seems unlikely that vampire bats would be the only bat species to engage in prosocial behaviors, as fruit bats have been observed participating in cooperative behaviors in the wild, such as mate guarding [6] and pup guarding [4]. A review of the existing literature suggests that kinship and the opportunity for reciprocity may be important factors in prosocial behavior [3]. Although our subjects consisted of close group mates, the extent of relatedness may have been less than in wild groups given the artificial means of group selection. Furthermore, given the housing conditions at OBC, we were not able to test mixed sex groups, and sex may also be an important factor in prosocial behavior. 

## 5. Conclusions

In conclusion, the results from this study revealed that, donor bats decreased the behavior of releasing the trap door of compartment B in both conditions over time, with a slightly sharper decrease taking place in the RA condition, while continuing to release the trap door at a similar frequency over time when a recipient bat was present. However, one documented failure to find evidence of prosocial behavior in Jamaican fruit bats should not dissuade future research efforts. Researchers should examine other contexts that would motivate bats to participate in data collection, and include the recording and possible manipulation of distress calls. The current procedure and findings should help point the way to new directions for exploring this topic in this, and closely related species.

## Figures and Tables

**Figure 1 behavsci-06-00025-f001:**
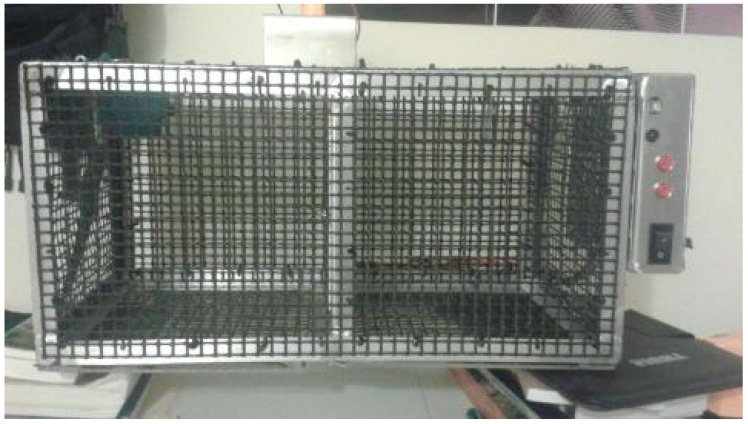
Apparatus in which bats were placed. The apparatus contained two compartments connected by an inner mesh door. Compartment A, on the left, contains the sensor in the top left corner. The bottom of each compartment contained a Lexan door, which, upon activation, opened to release the bat. The right control panel consisted of a USB port, two push buttons that, when pressed, released a trap door, and on/off switch. Resting on top of the apparatus was a DeWalt 18V XRP battery, which acted as the power source.

**Figure 2 behavsci-06-00025-f002:**
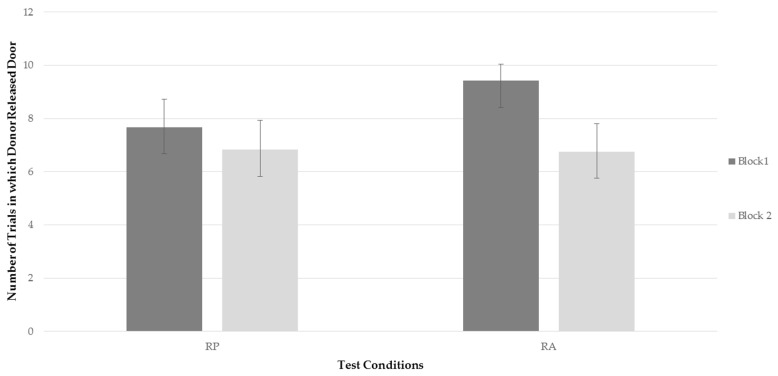
Average frequency for all bats to release the trap door of the recipient’s compartment in both test conditions within each testing block. Standard error represents the variability among bats within each condition/block.

**Figure 3 behavsci-06-00025-f003:**
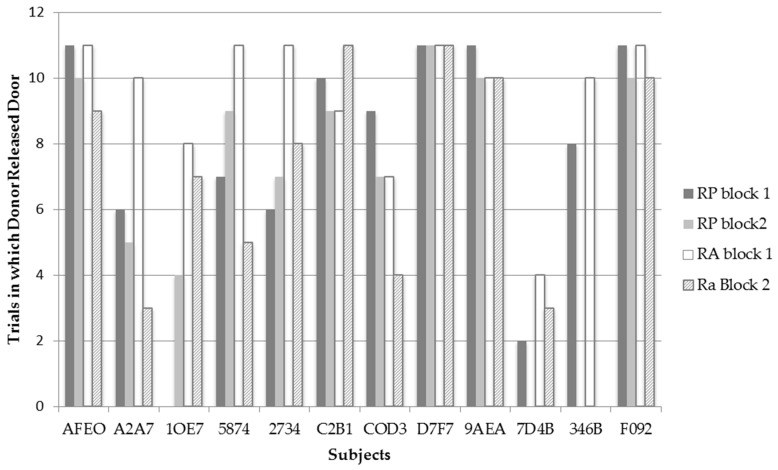
Frequency of each individual bat to release the trap door of the recipient’s compartment in both test conditions within each testing block.

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
