# Peer review of "Dynamic Duos? Jamaican Fruit Bats (Artibeus jamaicensis) Do Not Show Prosocial Behavior in a Release Paradigm"

_behavsci, 2016, doi:10.3390/bs6040025_

Round 1

Reviewer 1 Report

This is an important and valuable contribution to our understanding of the phylogenetic range of pro-social behavior. The basic result is that there is no evidence that a fruit bat (Artibeus jamaicensis) shows altruistic rescue behavior. Of course, as the authors note, this does not prove that this species of bat does not show any type of pro-social behavior but it adds to growing evidence that not all animals show all forms of pro-social behavior. Of note, the lack of altruism is in a mammalian species that is closely related to a species (the vampire bat) that does show prominent pro-social behavior.

The title is misleading. It is clearly meant to imply that pro-social behavior has been examined in the fruit bat but can easily be misconstrued to indicate that pro-social behavior has been observed in the fruit bat. For clarity, this title should be changed to reflect the finding.

There are several confusing aspects to the description of the methods; see details below.

I take issue with aspects of the framework used. The concepts of cost and gain appear to be economically defined. From a neurobiological point of view, internal reward is a gain and all movement, however trivial, incolces cost. Furthermore, the sentence in 354-5 suggests that a rat who acts to reduce her own distress is not acting altruistically. This has been addressed by Frans de Waal in his Ann Review Psych 2008 paper. Essentially de Waal argues compellingly that evolution has yoked the helper’s and victim’s affective states. This alternative idea should be at least acknowledged. This reviewer asks simply that the authors consider these points.

Lines 359-60: What does it mean that the species is very social?

Given that only two pregnant bats were investigated, this conclusion should be softened or omitted.

Minor

There is no reason to detail the number and housing of male bats (98-100).

The two parts of the mother-daughter sentence (100-101) should be made parallel.

The number of significant numbers used to give the arena dimensions is overkill. Round to nearest cm or at least nearest tenth.

What is a Lexan door?

In lines 119-120, it is made clear that there is no trap door in A nor any sensor in B. A is opened only manual. But this is contradicted in lines 125-6.

Lines 140 vs 144. 140 suggests that bats enter A on their own as during the habituation phase but it is clear from 144, bats are placed in compartment A.

First and second test blocks are talked about (168) long before they are introduced (205-6).

175-6: If all trials are 5 minutes in length then this is not a “maximum of 5 min.” It simply is 5 min.

What was the interval between RP trials with switched donor-recipient roles?

The sentence on 183-5 is unclear. Please rewrite.

316-7: This sentence can be understood but only after several reads.

345: it is not true that rats were unable to move in the restrainer in [10]. In fact rats could turn around in the restrainer.

Author Response

Thank you for the many helpful comments on our manuscript. Our responses to the reviewers, whose comments are italicized appear below.

Reviewer 1:

This is an important and valuable contribution to our understanding of the phylogenetic range of pro-social behavior. The basic result is that there is no evidence that a fruit bat (Artibeus jamaicensis) shows altruistic rescue behavior. Of course, as the authors note, this does not prove that this species of bat does not show any type of pro-social behavior but it adds to growing evidence that not all animals show all forms of pro-social behavior. Of note, the lack of altruism is in a mammalian species that is closely related to a species (the vampire bat) that does show prominent pro-social behavior.

Thank you for your positive comments on our manuscript.

The title is misleading. It is clearly meant to imply that pro-social behavior has been examined in the fruit bat but can easily be misconstrued to indicate that pro-social behavior has been observed in the fruit bat. For clarity, this title should be changed to reflect the finding.

We have changed the title to, “Dynamic Duos? Jamaican Fruit Bats (Artibeus jamaicensis) do not Show Prosocial Behavior in a Release Paradigm

I take issue with aspects of the framework used. The concepts of cost and gain appear to be economically defined. From a neurobiological point of view, internal reward is a gain and all movement, however trivial, incolces cost. Furthermore, the sentence in 354-5 suggests that a rat who acts to reduce her own distress is not acting altruistically. This has been addressed by Frans de Waal in his Ann Review Psych 2008 paper. Essentially de Waal argues compellingly that evolution has yoked the helper’s and victim’s affective states. This alternative idea should be at least acknowledged. This reviewer asks simply that the authors consider these points.

We would argue that acting to reduce one’s own distress does not meet the criteria of a prosocial motivation given that one’s behavior is then motivated to assist the self, not the other individual that is experiencing the distress. Although we agree that the affective states are connected in that an individual probably needs to experience the others’ distress in order to feel motivated to help another, other authors make a clear distinction between sympathy and empathy. Empathy doesn’t necessarily produce a sympathetic response. That is, I can detect your distress, but do I care about it? If I care only about reducing my own discomfort (e.g., stopping a crying baby from screaming because stressful sounds are aversive to me), this is not evidence that I have concern for the baby. We have referenced the suggested paper (thank you!) in acknowledging the challenges in this area.

Lines 359-60: What does it mean that the species is very social?

We have revised the sentence to be more specific, “As Jamaican fruit bats live in stable groups of several females with one or two males and spend a significant amount of time in close contact (Ortega & Arita, 2000), the current study then used an escape paradigm to test for prosocial behavior in females…”

Given that only two pregnant bats were investigated, this conclusion should be softened or omitted.

We’ve revised as follows, “; however our analysis revealed that donor bats were not more likely to act prosocially toward the two pregnant recipients in the current study.”

There is no reason to detail the number and housing of male bats (98-100).

We have deleted those sentences.

The two parts of the mother-daughter sentence (100-101) should be made parallel.

Good idea. We have changed the sentence as suggested, “Bat F092 was the mother of bat 346B and bat AF86 was the mother of bat 2734.

The number of significant numbers used to give the arena dimensions is overkill. Round to nearest cm or at least nearest tenth.

We have rounded the dimensions.

What is a Lexan door?

Lexan is a type of Plexiglas. We have changed to a “door made of Lexan”.

In lines 119-120, it is made clear that there is no trap door in A nor any sensor in B. A is opened only manual. But this is contradicted in lines 125-6.

We apologize if this is unclear but there is no contradiction here. Both trap doors could be opened using a button but the trap door in A could not be tripped by the bat via a sensor because there was a sensor available to the bats only in A.

Lines 140 vs 144. 140 suggests that bats enter A on their own as during the habituation phase but it is clear from 144, bats are placed in compartment A.

We have deleted the phrase “enter compartment A” from what was line 140 (now line 138)

First and second test blocks are talked about (168) long before they are introduced (205-6).

We have moved the sentences, “…, in the second block of trials a small cloth was hung on the side of the sensor that the bat had to go under or around in order to activate the sensor. The addition of the cloth further assisted the bat in determining when it was in the donor role versus the recipient role.” to the section relevant to the changes in Block 2.

175-6: If all trials are 5 minutes in length then this is not a “maximum of 5 min.” It simply is 5 min.

Good catch – thank you! We’ve removed “maximum of”

What was the interval between RP trials with switched donor-recipient roles?

RP trials were conducted back to back with the donor and recipient switching roles.

The sentence on 183-5 is unclear. Please rewrite.

We have deleted the sentence.

316-7: This sentence can be understood but only after several reads.

We have rewritten that sentence.

345: it is not true that rats were unable to move in the restrainer in [10]. In fact rats could turn around in the restrainer.

Thanks for this correction. We have revised as, “In [10], the recipient rats were constrained in a manner that may have caused high levels of distress that donor rats could presumably detect, which might have motivated them to release their trapped cage mates.”

Reviewer 2 Report

Cooperative behavior is a fascinating topic, and I think readers of this journal would be very interested in this study, but some revisions are needed. The authors tested the ability of fruit bats to learn to release themselves from a cage and then to release another bat from a cage. They found no evidence that the bats released another bat from the cage. However, the paper does not present an analysis clearly demonstrating whether or not the bats actually learned to release themselves. So it was not clear to me if the bats understood how the apparatus works.The paper as written also does not show that the second bat was in distress. Finally, the statistical model (a parametric test) assumes that the response data came from a normal distribution, but there is no evidence given for this. The authors should test whether the residuals from their model are normally distributed. Parametric tests should not be applied to response data that are zero-inflated, skewed, or have outliers. Alternatively, rather than changing the analysis, the authors can simply publish their data for others to inspect more thoroughly. After these revisions, I would like to see this work published. There is often a publication bias against negative results of this kind, and of course they are of great scientific importance.

Comments by line

Title:

The bat’s genus name is Artibeus; it is spelled wrong in the title and abstract.

The title might be misleading since they found no evidence of prosocial behavior.

9 Who thinks prosocial behavior is uniquely human? Please cite.

13 Why is it that diet is expected to impose different selection pressures and not something else more directly related to social behavior?

18 present and absent on four occasions each? or present twice and absent twice?

19 decreased their propensity to open the trapdoor over time?

27-28 According to this definition of prosocial, all animals with some parental care show prosocial behavior. All eusocial insects and cooperatively breeding birds and mammals perform prosocial behavior beyond offspring. Therefore it seems to me obvious that many species perform prosocial behavior. Please clarify.

37 Define “microbat” — this is not a monophyletic group. My understanding is that notion of micro- and megachiroptera is outdated from an evolutionary perspective.

39 Reference 3 does not support this statement.

41. Why is it not prosocial? If bats behave to assist others, then it clearly fits the definition given earlier. What is the behavior? Huddling together?

50 extra parenthesis

51. period missing

52. What makes it “true” prosocial behavior? What is “false” prosocial behavior?

56, 58 indirect fitness benefits

65 “controlled” setting implies that the tests with primates and rats are identical, which is not true

67 define controlled setting. Do you just mean captive?

72 Not all bats form close social bonds beyond mother-offspring.

88-89 This part is confusing. Are they releasing themselves or the other bat?

95 add species name

101 change to “Relatedness was unknown…”

106 Institutional Animal Care and Use Committee

114 So to open the trapdoor, the bat had to go in the corner near the door or away from the door?

138 Is the purpose of Phase 1 to train the bats to release themselves?

149 How quickly did the bats have to exit to reach criterion.

188 unmotivated to do what?

statistics

Were residuals normally distributed? Were the variances equal?

Why is time not being tested as a continuous variable to get the linear slope of behavior rate over time?

Figure 2. It would be much better to see the actual distribution of data. The error bars are not explained. 

I would like to see an analysis of the training results to assess the evidence that the bats learned how to release themselves and did not just do so by chance (wandering around the cage).

292 In order to demonstrate prosocial motivations, the animal first must understand how to operate the apparatus in order to use it to help the recipient. What is the evidence that the bats 1) understood the apparatus and 2) understood they could help another bat, and 3) that the recipient was distressed.

379 vampire bats do not need to feed every day

382 vampire bats do not (I think) have larger relative brain sizes than some flying foxes

392 Why would relatedness be less in captivity? It seems that it would be actually greater than in nature due to with inbreeding and lack of dispersal.

395 I would delete this section. You cannot draw such conclusions from just these 2 cases.

The authors should provide their raw data in a supplement for further analysis or meta-analysis.

The references have extra template text that needs to be deleted.

Author Response

Cooperative behavior is a fascinating topic, and I think readers of this journal would be very interested in this study, but some revisions are needed. The authors tested the ability of fruit bats to learn to release themselves from a cage and then to release another bat from a cage. They found no evidence that the bats released another bat from the cage. However, the paper does not present an analysis clearly demonstrating whether or not the bats actually learned to release themselves. So it was not clear to me if the bats understood how the apparatus works.The paper as written also does not show that the second bat was in distress. Finally, the statistical model (a parametric test) assumes that the response data came from a normal distribution, but there is no evidence given for this. The authors should test whether the residuals from their model are normally distributed. Parametric tests should not be applied to response data that are zero-inflated, skewed, or have outliers. Alternatively, rather than changing the analysis, the authors can simply publish their data for others to inspect more thoroughly. After these revisions, I would like to see this work published. There is often a publication bias against negative results of this kind, and of course they are of great scientific importance.

Thank you for your encouragement to publish these null results. Also, thank you for the comment regarding normality. We have in fact inspected the data and determined them to be normally distributed.

This reviewer raises an important point about whether the bats had really learned how the apparatus worked. It is possible that they did not behave prosocially because they did not understand the consequence of their action of moving in compartment A to releasing the door in compartment B. We have emphasized that possibility more heavily now in the discussion.

The bat’s genus name is Artibeus; it is spelled wrong in the title and abstract.

Thank you so much for catching that error! We have corrected it now.

The title might be misleading since they found no evidence of prosocial behavior.

We have changed the title.

Who thinks prosocial behavior is uniquely human? Please cite.

It is not typical to provide citations in an abstract. The idea that prosocial sentiments are unique to humans has been expressed prolifically in the comparative psychology literature.

13 Why is it that diet is expected to impose different selection pressures and not something else more directly related to social behavior?

We are working from the framework of the technical intelligence hypothesis, which posits that foraging pressures may exert a selective force on cognitive capacities, such as memory, the ability to track events etc. Such abilities may also assist in acting on prosocial motivations, by allowing animals to keep track of conspecifics’ actions, for example. We have added, “given that prosocial preferences may depend upon cognitive abilities selected by different ecological constraints” to the abstract.

18 present and absent on four occasions each? or present twice and absent twice?

Present and absent on two occasions each. This has been revised.

19 decreased their propensity to open the trapdoor over time?

Yes, we have added the phrase “over time”.

27-28 According to this definition of prosocial, all animals with some parental care show prosocial behavior. All eusocial insects and cooperatively breeding birds and mammals perform prosocial behavior beyond offspring. Therefore it seems to me obvious that many species perform prosocial behavior. Please clarify.

We do not consider parental care to be the type of behavior that results in “no gains to the self” given that increasing an offspring’s fitness increases one’s overall reproductive fitness. There is also a gain to the self if behaving prosocially results in reciprocated acts of reciprocity over time, which could explain many instances of apparent altruism, such as those you describe. We have added a few sentences to clarify.

37 Define “microbat” — this is not a monophyletic group. My understanding is that notion of micro- and megachiroptera is outdated from an evolutionary perspective.

Thank you for this. We have removed reference to this species as a “microbat”, and also to Jamaican fruit bats.

39 Reference 3 does not support this statement.

We have replaced this reference.

41. Why is it not prosocial? If bats behave to assist others, then it clearly fits the definition given earlier. What is the behavior? Huddling together?

We have explained that it is not strictly prosocial because the potential benefits outweigh any costs.

50 extra parenthesis

Thank you. We have deleted this.

51. period missing

Thank you. We have added a period.

52. What makes it “true” prosocial behavior? What is “false” prosocial behavior?

We have revised to read, “Although strictly prosocially motivated behavior”.

56, 58 indirect fitness benefits

Agreed, thank you!

65 “controlled” setting implies that the tests with primates and rats are identical, which is not true

67 define controlled setting. Do you just mean captive?

That is certainly not what we meant to imply. We meant controlled in the sense that systematic pairings and manipulations could be presented. We have changed to “controlled experimental design”. We have removed reference to “controlled setting”.

72 Not all bats form close social bonds beyond mother-offspring.

We have revised to, “Because bats can form close social bonds [3, 11- 12) across multiple dimensions,…”

88-89 This part is confusing. Are they releasing themselves or the other bat?

We have added, “to allow its partner to escape”

95 add species name

Done.

101 change to “Relatedness was unknown…”

This isn’t true. We did know relatedness along matrilines. We have revised.

106 Institutional Animal Care and Use Committee

Done.

114 So to open the trapdoor, the bat had to go in the corner near the door or away from the door?

We have indicated where the sensor was located in the caption for Figure 1.

138 Is the purpose of Phase 1 to train the bats to release themselves?

The purpose was to train them to learn the association between movement near the sensor and the opening of the trap door. It was also anticipated that they would be motivated to do so based on the experience of releasing themselves and that this behavior would dissipate only in the RA condition if prosocial, but if in both conditions if not prosocial similar to Vonk et al. (2008).

149 How quickly did the bats have to exit to reach criterion.

Any time within the 5 minute trial.

188 unmotivated to do what?

We have added “to move within the apparatus”

Were residuals normally distributed? Were the variances equal?

For the frequency data (release or not release), it does not make sense to examine normality. For the latency data, the data is skewed when we include the maximum trial length to replace trials on which the trop door was not released. There is no appropriate transformation for this issue. Thus, we elected to add an analysis dealing with only the data for which bats released the trap door on both RP and RA trials.  This data was normally distributed when viewed via histogram.

Why is time not being tested as a continuous variable to get the linear slope of behavior rate over time?

It is unclear what you are suggesting here. Time did not really vary as the bats were only tested across 22 total test days. The change was from one session to the next, not from a specific increment of time to another. Latencies to release were compared as function of time block because there were only two trials in each condition to compare.

Figure 2. It would be much better to see the actual distribution of data. The error bars are not explained. 

It isn’t clear what the reviewer is suggesting here. We do present all of the individual results in Figure 3. Does the reviewer wish to see a Table of latency or release data instead of the figure presented? The figure is presented to explain the interaction that approached significance. We have added an explanation of the error bars.

I would like to see an analysis of the training results to assess the evidence that the bats learned how to release themselves and did not just do so by chance (wandering around the cage).

292 In order to demonstrate prosocial motivations, the animal first must understand how to operate the apparatus in order to use it to help the recipient. What is the evidence that the bats 1) understood the apparatus and 2) understood they could help another bat, and 3) that the recipient was distressed.

It is possible that the bats did learn by chance. Regardless, the fact that they activated the sensor allowed for the possibility that that action would become associated with the expectation of the trap door opening in chamber B. The key prediction was that bats would continue to release the trap door indiscriminately across conditions if the behavior was random, but would do so only in the RP condition if prosocial. Our data does not allow us to discriminate between these possibilities. We have addressed the possibility in the discussion that bats did not learn the association adequately to express prosocial sentiments.

379 vampire bats do not need to feed every day

Thank you. We have corrected.

382 vampire bats do not (I think) have larger relative brain sizes than some flying foxes

We have added “among” to qualify the conclusion just in case.

392 Why would relatedness be less in captivity? It seems that it would be actually greater than in nature due to with inbreeding and lack of dispersal.

We added “given the artificial means of group selection”

395 I would delete this section. You cannot draw such conclusions from just these 2 cases.

Agreed. We have deleted it.

The authors should provide their raw data in a supplement for further analysis or meta-analysis.

We are happy to provide the data file upon request. However, based on our MOU with the Organization for Bat Conservation, we would need their permission before providing the data openly.

The references have extra template text that needs to be deleted.

Thank you for catching this!